**Review article**

# Genetic drivers of congenital cardiac fibrosis
Angela C. Zeigler [1,2] & Marlin Touma [1,2] ✉

Cardiac fibrosis in congenital heart disease (CHD) is associated with poor outcomes, but the genetic risk factors have not been clearly outlined. This review details genes important for regulation of normal cardiac development or fibrosis, particularly cilia-related genes. Specific CHD pathologies have different patterns of fibrosis, likely from interaction between genetic mutations and environmental factors. Future studies are more feasible as tools like single-cell RNAseq and patient-derived organoids have become more affordable and easier to implement. A better understanding of genetic risk factors for fibrosis in CHD could improve diagnosis and treatment for these patients.

Congenital heart disease (CHD) affects 6–10 per 1000 live births each year[1]. The prevalence is increasing in adults and children because of advances in cardiothoracic surgery and post-surgical management[1]. In CHD, the morphology of the heart is abnormal at the gross, tissue, and cellular levels. Cardiomyocytes (CM) often have associated pathological characteristics in CHD[2,3]. However, cardiac fibroblasts (CF) and endothelial cells (EC) also show abnormal patterns of proliferation, activation, and extracellular matrix (ECM) production in CHD[4–8]. Pathological ECM composition and expansion contributes to the morbidity and mortality of infants with CHD because it is associated with increased risk for arrhythmia and heart failure[6,9,10].

Not all patients with CHD have an identified genetic cause, and many patients have a constellation of mutations in genes associated with cardiomyocyte differentiation, sarcomere structure, fibroblast activation, cellular proliferation, and ECM regulation[11,12]. Only 20–45% of patients with CHD have a clear genetic cause including aneuploidies, copy number variants, and monogenic mutations—generally falling under the umbrella of syndromic CHD[12,13]. Aside from pathogenic genetic variations, altered gene expression patterns due to epigenetic changes or changes in micoRNA expression have also been linked to CHD pathogenesis[14,15]. The search for genetic causes for congenital cardiac pathologies generally focuses on the type of structural anomaly. In general, few studies have linked genetic predisposition with fibrosis patterns in CHD. However, alterations in ECM protein expression or mutations in ECM-associated proteins have been associated with increased risk of CHD[3,11,16].

The genetic basis for fibrosis in CHD has been particularly difficult to study. There are several reasons leading to this difficulty, including a paucity of human samples, inability to translate fibroblast data from mice or rats to humans, and the slow progression of fibrosis making it unclear what is a response to e.g. surgical injury versus genetically predisposed[6,17–20]. Few studies have related cardiac fibrosis to pathogenic genetic variants or altered gene expression[3,21]. Further, the outcome of cardiac fibrosis is incredibly heterogenous encompassing different patterns and total area of ECM, and variable compositions[6,8,19]. Therefore, the genetic and environmental factors that put a CHD patient at risk for cardiac fibrosis are unclear. However, the rise of technologies, such as single cell RNA sequencing (scRNA-seq), advanced cardiac imaging, and discovery of circulating biomarkers have opened the door to further identification of genes and signaling pathways that are involved in cardiac fibrosis[20–23]. Among the advantages of scRNA-seq are the ability to directly study human cardiac tissue from biopsies or explants and the feasibility of examining different cell types concurrently[22,24]. The use of patient-derived induced pluripotent stem cells (iPSCs) and generation of cardiac organoids from these cells combined with scRNA-seq has opened the door to very detailed mechanistic studies relating organoid function to expression patterns in the context of genetic variation[25–27]. Advanced cardiac imaging has allowed us to characterize fibrosis in living patients when access to histological data is unreasonable[9,21]. Validation of circulating biomarkers in humans allows for easily tracking fibrosis development over time which will be important for further classifying this serious pathology[28,29].

Understanding the risk factors for developing cardiac fibrosis is important. We know that the presence of cardiac fibrosis predicts morbidity and mortality[19,23]. Relevant genetic testing could allow for early recognition of patients at risk for developing fibrosis.

Further, a more thorough knowledge of the genetic factors could also point to potential therapeutic strategies particularly aimed at preventing the development of cardiac fibrosis which is generally irreversible[23].

Here we present an overview of the current understanding of the signaling pathways and genes involved in fibrosis of CHD. We will outline the origin for fibroblasts and mechanisms of normal ECM production during cardiac development including the unique role of cilia and the regulation of EndoMT and EpiMT. Specific structural heart diseases and their

[1]UCLA David Geffen School of Medicine, Neonatal Congenital Heart Laboratory, 675 Charles E. Young Drive, MRL Building Laboratory 3447/3457, Los Angeles, CA, USA. [2]UCLA David Geffen School of Medicine, Department of Pediatrics, Marion Davies Children's Center 10833 Le Conte Ave, Los Angeles, CA, USA. ✉e-mail: mtouma@mednet.ucla.edu

known associated fibrosis patterns are discussed in detail along with the genes linked to CHD that likely affect fibrosis risk. Although this is an area of research that has many remaining questions, improvements in clinical measurements of fibrosis, tissue engineering technologies, and computational techniques will enable future studies to tackle the questions raised in this review.

## Discussion

### Fibroblasts and ECM production during normal heart development

The heart is one of the first organs to develop during embryogenesis generally being completed by day 40 of gestation (Fig. 1)[13]. In normal cardiac development, the first heart field (lateral plate mesoderm) and second heart field (lateral plate splanchnic mesoderm) coalesce into a crescent shape. At this point, signals, such as FGF and BMP commit these cells to a cardiac fate (particularly to CM)[30]. Neural crest cells are also incorporated to form the primitive heart tube[31,32]. The looping of the cardiac tube is completed by around 30 days of gestation[2,33]. Abnormalities in looping are thought to arise from issues in left-right patterning associated with genes such as *HAPLN1, NODAL, LEFTY1*, and *LEFTY2* and with ciliary function which is described in detail below[32,34]. Abnormal looping can result in a variety of pathologic phenotypes including corrected transposition of the great arteries, tricuspid atresia, heterotaxy, and double-outlet right ventricle. At around 4-6 weeks of gestation, the endocardial cushions and conal / truncal cushions form to create four ventricles and two separate great vessels (the aorta and main pulmonary artery). *Tie2* and *CDH5* are two of the genes shown to be important for cushion formation[31]. These cushions are formed from the migration of neural crest cells and fibroblasts or proto-fibroblasts. Abnormal cushion formation can lead to CHD phenotypes, such as truncus arteriosus, arterioventricular septal defects (AVSDs), atrial septal defects (ASD), ventricular septal defects (VSD), transposition of the great arteries, and tetralogy of Fallot (ToF)[32].

Fibroblasts are remarkably heterogeneous, which has made accurately identifying them and the pathways that regulate them during development traditionally difficult[29,31,35]. However, lineage tracing studies have made it easier to follow the development of cardiac fibroblasts from endocardium, epicardium, and neural crest cells. In mice, fibroblasts are not initially present in the developing heart, but after the formation of the right and left chambers fibroblasts are present in the heart and increase in number until the neonatal period[18]. The majority of fibroblasts arise from epicardial origin via epithelial to mesenchymal transition (EpiMT), and these cells are WT1 and Tcf21 positive in mice[31,36]. Mouse studies have shown that PDGFRa, WT1, and Tcf21 signaling are all important for the EpiMT process, and those proteins are used as markers of post-EpiMT fibroblasts [Supplementary Data 1][37]. Endothelial to mesenchymal transition (EndoMT) is the process by which the remainder of fibroblasts are generated in the heart. Neural crest cells give rise to mesenchymal stem cells (and ultimately fibroblasts) through EndoMT, which tend to migrate to the outflow tracts and valves[31,38]. Thus, the endocardial cushions and valves are populated with fibroblasts from EndoMT, and these cells are positive for endothelial markers, such as Tie2 and VE-cadherin in mice[31,37].

Several signaling pathways have been identified that regulate cardiac fibroblasts behavior during development. FGF10 signaling has been shown to be important for migration of fibroblasts into the myocardium[31]. EndoMT during formation of the endocardial cushions is induced by signaling molecules that are upregulated during cardiac development including SNAIL, TGFB, NFATc1, and SOX9[39]. In contrast, BMP signaling can decrease EndoMT in endothelial cells thereby regulating fibroblast differentiation[17]. As the technology around scRNA-seq has improved, new studies have more clearly delineated different populations of cardiac

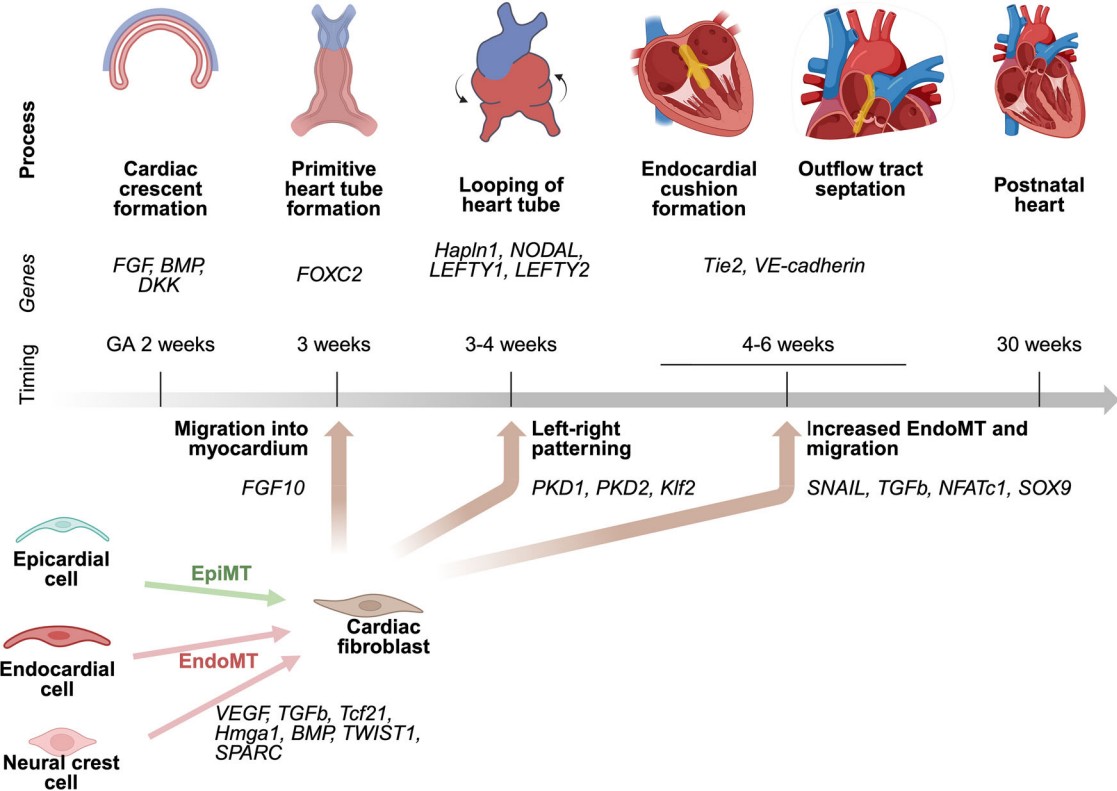

**Fig. 1 | Major events in cardiac development.** Each major event is shown with the approximate timing during gestation and key genes known to affect that phase of development and have a role in regulating fibroblast behavior. On the bottom, fibroblasts are shown differentiating from epicardial, endocardial, and neural crest cells then participating in various steps of development. Key genes that have been shown to regulate each fibroblast-associated process are indicated next to the brown arrows. *Created in BioRender. Zeigler, A. (2026)* https://BioRender.com/uldi9uo.

fibroblasts and their activity during cardiogenesis. Performing scRNA-seq on human embryonic cardiac isolates has specifically highlighted different expression profiles at different stages of cardiac development[17,40]. Initially, fibroblasts express genes that attenuate pro-fibrotic signaling and are more closely related to proliferation and increased metabolism, such as ribosomal proteins, COX7C, ELOB, and BAMBI. Later in development, fibroblasts express genes related to extracellular matrix production especially COL1A1, COL3A1, and FBLN1[17,40]. Proliferative capability of fibroblasts decreases throughout development, but the fibroblast population can increase in the neonatal period or in response to injury in adults[17].

The spatial orientation of the heart is particularly important for normal structural development. Differences in gene expression in different regions of the heart are established early on in development and persist thereafter[2]. Fibroblasts with different expression profiles cluster in the endocardial cushions / valves, the outflow tracts, and in the ventricular walls around blood vessels[2]. In adults, distinct populations of fibroblasts have also been found in the atria vs the ventricles[24]. The ventricles have a higher number of fibroblasts that express genes responsive to TGFB signaling or are responsible for ECM remodeling[24]. The same sort of spatial specificity has been established for CM, endothelial cells, and other cell types[2,24,41]. One important mediator of regional specificity are cilia which are present on cardiac progenitor cells as well as cardiac fibroblasts[12]. Primary cilia coordinate signaling pathways, such as Nodal, Shh, Wnt, and Pkd, which are important for left-right patterning[33,38,42,43]. Motile cilia are necessary for normal cell migration during development which is important in various parts of cardiogenesis - in particular the formation of the endocardial cushions and the outflow tracts[42,44].

Fibroblasts play important roles in regulating the maturation of cardiac tissue and the differentiation of CM. Mechanical signals via integrins, fibronectin, and periostin are generated by fibroblasts and stimulate proliferation and differentiation of CM[31,45]. As the pressure increases on cardiac tissue from blood flow, the ECM scaffold is reinforced by fibroblasts primarily with collagen I (COL1A1), collagen III (COL3A1), and fibronectin (FN1). The stiffening of ECM is important for the maturation of sarcomeres by regulating the orientation and terminal differentiation of CM[45]. As the tissue matures, fibroblasts continue to support myocardial tissue through maintenance of the ECM. They also participate in direct cell-cell signaling and electrical conduction through connexins proteins that assemble to form gap junctions connecting them to CM[29,45]. The details regarding the role of mesodermal cells during cardiac development and the differentiation of CM is outside the scope of this review and is reviewed elsewhere[13,30,46]. However, it is worth noting that because fibroblasts play a role in cardiomyocyte differentiation, they likely play a role in the structural aspects of CHD in addition to the risk for fibrosis.

## Markers of cardiac cell types and cardiac fibrosis
Cardiac fibroblasts have been somewhat difficult to define because they are remarkably plastic and heterogenous making any marker used to identify them either non-specific, thereby marking other cell types, or too specific, therefore only capturing a subpopulation[2,3,24,31,47–49]. It is particularly difficult to track fibroblasts during development because they derive from multiple different cardiac progenitor cells, as described above, and will often retain markers of the original cell type for some period of time[31,37]. In Supplementary Data 1, we summarize some of the most common markers for cardiac fibroblasts during development, quiescent maturation, or response to injury (activation), as well as their overlap.

## Role of cilia in cardiac development and fibrosis
Cilia are membrane bound microtubules that protrude from the cellular surface and coordinate external chemical and mechanical stimuli to formulate specific cellular response mechanisms[50,51]. The 9 + 0 (primary cilia) or 9 + 2 (motile cilia) axoneme pattern is connected to a basal body and modified centriole at the base, and it's connected to the surrounding membrane by transition fibers which regulate transport of proteins into the cilia at the transition zone[33,38,50]. The ciliary membrane is enriched with

channels and receptors that serve to facilitate cross-talk between competing signals[51]. Protein complexes form the intraflagellar transport (IFT) trains driven by kinesin and dynein proteins, and these carry necessary microtubule building blocks, translation machinery, and signaling proteins across the transition zone and to the tip of the cilia[52]. Overall, this structure creates a highly regulated microenvironment for sensing external stimuli and coordinating responses within the cell [Fig. 2].

Cilia play important roles during development and are present on most cell types in the developing heart, whereas they are primarily observed in the adult heart on CF during wound healing[50,51,53]. Left-right organizers (LRO) are transient nodes of tissue that appear during development and create asymmetrical extracellular fluid flow through motile cilia that triggers the development of normal asymmetry as the flow is sensed by other cells through their primary cilia[51,54]. Effective LRO signaling is one of the regulators of normal cardiac dextral looping (see Fig. 1). Therefore, both motile and primary cilia are necessary for normal left-right differentiation and the looping stage of cardiac development[13,33,55]. Mutations in genes related to ciliary structure have been linked to heterotaxy in humans, such as DNAH11 and DNAH5 [Supplementary Data 2][50,56].

Migration of progenitor cells during development is also dependent on normal cilia function. Sonic hedgehog (Shh) signaling, mediated by cilia, is particularly important during migration into the endocardial cushions and septation of the outflow tract. Multiple genes including Msk1, Tctn2, and Ftm have been shown to be important for normal Shh signaling and ciliary function, and mutations in these genes are linked to atrioventricular septal defects[51]. A VEGF associated gene, CRELD1, that is important for primary ciliary structure was linked to AVSDs in humans even independently from heterotaxy, and newer studies have linked signaling in primary cilia with AVSD[50]. Primary cilia sense fluid shear stress, and there is evidence this process is important for normal septation and ventricular structure[51]. Mechanosensing proteins on primary cilia, Pkd1 and Pkd2, regulate calcium signaling and the transcription factor Klf2 which seems to be an important process for triggering EndoMT in order to populate the endocardial cushions[50,51,57]. Primary cilia facing into cardiac chambers have been observed in mice and zebrafish, and this could be an explanatory mechanism for how sheer stress from blood flow regulates growth of the cardiac chambers and great vessels[50].

Cilia coordinate various signaling pathways that are important for cardiac progenitor cell and cardiac fibroblast function, such as Shh, TGFB / BNP, Wnt, PDGF, and NOTCH signaling[38,43]. As mentioned above, Shh signaling is mediated by cilia and facilitates normal migration particularly during endocardial cushion formation. TGFB receptors are concentrated at the base of cilia which facilitates downstream smad activation. TGFB / smad signaling is essential for EndoMT and ECM production particularly during endocardial cushion and valve formation[33,38]. Noncannonical Wnt signaling components, such as Inversin and Wdpcp have been observed to be concentrated in cilia structures, and mutations in Wdpcp specifically have been shown in mice to lead to shortened outflow tract development likely from impaired directional migration[38]. PDGFRa localizes to cilia, and PDGFRa-mediated signaling is important for migration and proliferation particularly with respect to ventricular development[33]. Notch signaling is regulated by intraflagellar transport and mechanosensation in primary cilia, and members of the notch pathways including Hrt1, Hrt2, and Hrt3 have been shown to be important for EndoMT and normal CM differentiation[33,58,59].

It's clear there is a genetic link between mutations in cilia-related genes and CHD. Multiple ciliopathies have been associated with cardiac abnormalities in addition to the primarily affected organs[12]. Cilia-related genetic mutations have been associated with CHD in mice, most notably in a study by Li et al. where they found mutations in ciliary genes enriched in mice with CHD phenotypes using a large panel of mouse mutants[44,60]. Many of those mutations have also been associated with CHD in humans[38,43,44]. As mentioned above, heterotaxy and AVSD are particularly linked to ciliary defects because of the roles of cilia in cellular migration and left-right patterning[44,51]. However, EndoMT and normal endocardial cushion / valve

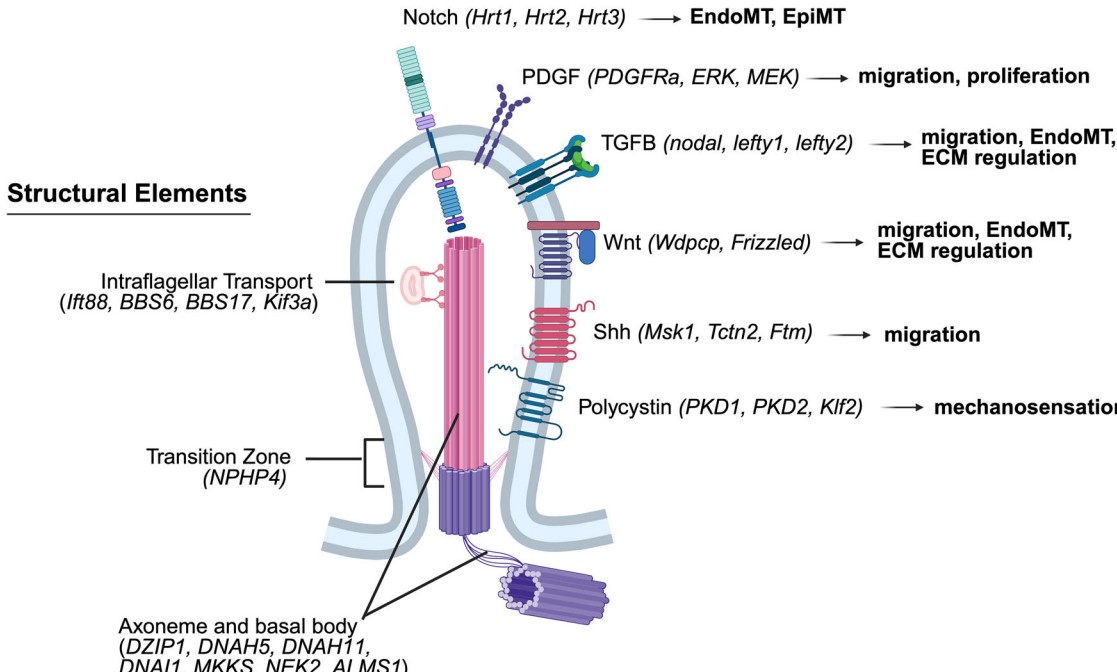

**Fig. 2 | Cilia structure and signaling regulation.** The general structural elements of the cilium including the intraflagellar transport, transition zone, and microtubule structures (axoneme and basal body) are indicated along with key structural genes in parentheses that are associated with congenital heart disease (CHD). On the left are receptors for major signaling pathways and the cellular functions they are associated with in cardiac fibroblasts during normal development. The pathway-associated genes indicated are those with a likely role in fibrosis and in which mutations have been linked to CHD. *Created in BioRender. Zeigler, A. (2026)* https://BioRender.com/tlkwh58.

formation has also been linked to cilia through coordination of TGFB and Shh signaling[33,50,51].

Cilia certainly regulate cardiac fibroblast activity during development and in the adult heart, so there likely is a connection between mutations in ciliary proteins and fibrosis in CHD. Several ciliopathies with known genetic mutations have been linked to fibrosis in other organs[12]. The strongest link between ciliary dysfunction and pathological cardiac fibrosis is the identification of a likely causal loss-of-function mutation in *ALMS1*, a protein important for cilia formation, in a patient with primary endocardial fibroelastosis (EFE)[61]. Fibroblasts with cilia have been observed in adult human hearts with heart failure and fibrosis[53]. Rat studies have shown that cilia-related proteins, such as polycystin 1 (*Pkd1*) are important in mediating a pro-fibrotic response to injury likely through a TGFB-dependent pathway[50,53]. In fact, many pathways related to fibroblast activation have been shown to be dependent on normal cilia function including TGFB, PDGF, and Wnt signaling, although this cilia-dependent signaling has primarily been studied with respect to left-right patterning rather than fibrosis, meaning that link has not been definitively shown[13,38,43]. Cilia-dependent signaling pathways have also been linked to altered EpiMT or EndoMT indicating a role for regulating the size of the fibroblast population in the heart[33,38,50]. During development, cilia diminish in mitral valves as collagen is deposited, and loss of cilia altogether in valves leads to abnormal ECM deposition ultimately predisposing to valve prolapse[51]. Specifically, mutations in *DZIP1* (a gene that regulates ciliogenesis) and *Ift88* (an IFT-related gene) have been clearly associated with abnormal ECM activation that leads to mitral valve prolapse[51]. All of this together, is strong evidence that the ciliary mutations identified in Li et al's forward mice screen could also be linked to fibrosis in CHD rather than just the structural phenotype[60].

Importantly, cilia might be a key for understanding multi-genic risk for CHD and cardiac fibrosis. Cilia coordinate signaling through spatial organization of signaling proteins from many different pathways, and Li's mouse genetic screen found several groups of genetic mutations from different pathways that interact physically in cilia have a similar physiologic outcome. For example, mutations in *Bicc1*, *Anks6*, *Nek8*, and *Wwtr1* all cause outflow tract defects and are known to physically interact[43,60]. Interactome studies have also shown a connection between cilia-related proteins and proteins derived from genes linked to CHD hinting at a further role for cilia in mediating multigenic risk[60,62]. Future studies linking ciliary proteins to fibrosis-associated proteins or signaling pathways may highlight potential multi-genic risk factors for developing cardiac fibrosis in the setting of CHD.

## Cardiac fibroblasts in the pathology of congenital heart disease and fibrosis

CHD, as explained in more detail below, involves a spectrum of pathologies at the tissue level including malalignment of outflow tract tissues, valve abnormalities, vascular anomalies, CM hypertrophy, and fibrosis. As outlined above, cardiac fibroblasts play a prominent role in the development of the heart, and altered fibroblast activity during cardiac development plays a role in abnormal cardiogenesis. Normal left-right patterning and normal septation of the chambers of the heart and the great vessels all depend on normal fibroblast differentiation, migration, and ECM production (Fig. 1). There have been many studies investigating the genetic cause for the various types of CHD[11–13,63], and overall it seems there is generally a polygenic cause in patients with non-syndromic CHD. Supplementary Data 2 outlines several mutations strongly associated with CHD. Notably, many genes clearly involved in ECM production, such as *COL1A1*, *COL5A2*, *FBN2*, and *ELN* have been associated with CHD in whole-exome sequencing of patients with CHD[11–13]. Although these proteins do have some feedback signaling mechanisms on fibroblast function, these are likely downstream of the truly causal genetic mutations. Cardiomyocyte-specific genes have been linked to different CHDs including ones related to sarcomere structure (*MYBPC3*, *MYH6*)[11,13,27]. These genes were thought to be less likely to be directly

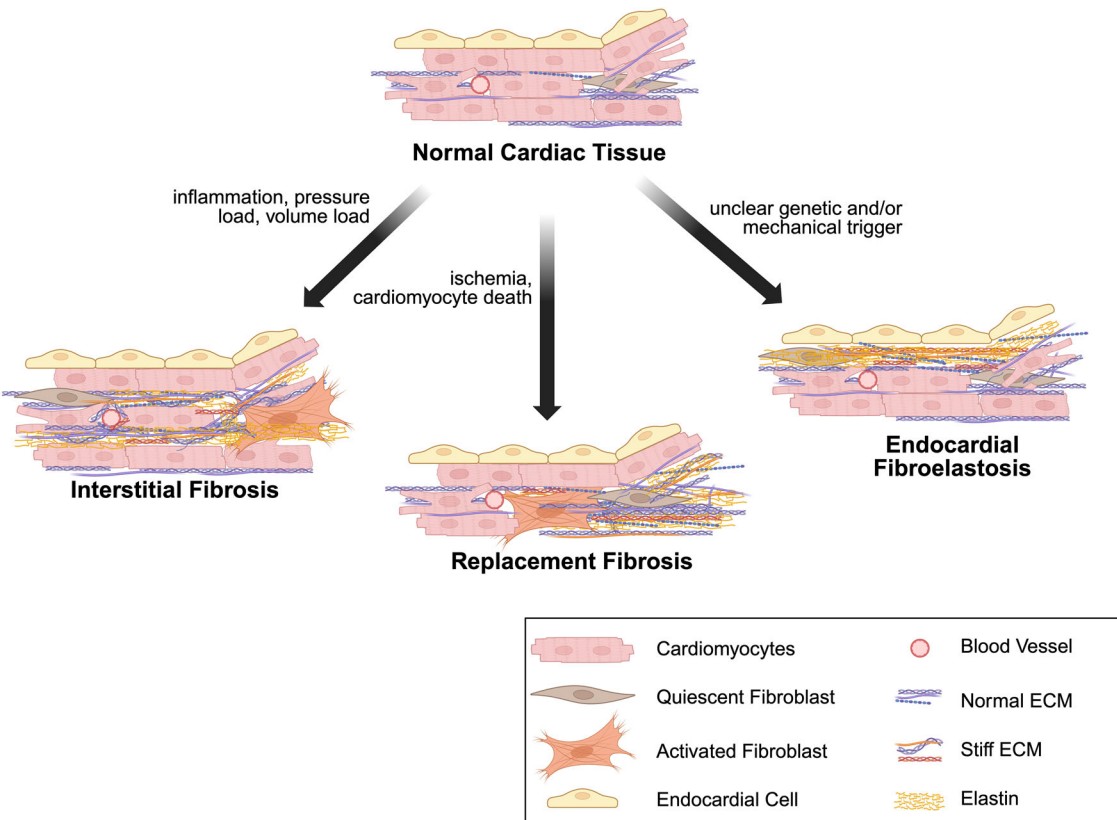

**Fig. 3 | Patterns of fibrosis.** Normal cardiac tissue consists of cardiomyocyte (CM) aligned with extracellular matrix (ECM) scaffold and quiescent cardiac fibroblast (CF). Interstitial fibrosis develops with no loss of CM but an increase in ECM deposition between CM and around blood vessels, addition of stiffer ECM proteins, and activation of fibroblasts. Replacement fibrosis occurs with loss of CM and replacement with stiff ECM and activated fibroblasts. Endocardial fibroelastosis is characterized by a layer of elastin rich deposition under the endocardial cells and an increase in CF in that area. Endomyocardial fibrosis (not depicted here) is similar to endocardial fibroelastosis in appearance but not related to CHD. Created in BioRender. Zeigler, A. (2026) https://BioRender.com/3yy6jlr.

involved in cardiac fibrosis, but cardiomyocyte-specific pathology has been shown to alter the ECM[21]. As mentioned above, ciliary defects are enriched in CHD (*DNAH11, DNAH5, PKD1L1*), and ciliary function is important for normal fibroblast formation and activation during development which could represent an important mechanistic link between CHD pathogenesis and the risk for cardiac fibrosis[12,53]. There are also CHD-associated genes related to transcription factors or signaling pathways that affect normal differentiation of many cell types including CM, CF, and EC, making their link to cardiac fibrosis specifically less clear (*GATA4, GATA6, NKX2-5, TBX5, NOTCH1, JAG1*)[13,64]. VEGF signaling members (*FLT4, KDR*) are important for normal EC function during development and are also strongly associated with CHD with unclear implications for fibrosis risk[13,63].

There are important differences between how fibroblasts behave in infant or adult hearts versus fetal or perinatal hearts, and this contributes to some of the unique patterns of fibrosis seen with CHD [Fig. 3]. Adult CFs play a major role in modulating the entire post-myocardial infarction wound healing process through chemokine attraction of immune cells, proliferation and migration to the site of injury, and formation of new ECM – often to replace nonviable CMs as described above[65]. This process can also be triggered by infection or abnormal wall tension. In fact, a study performing snRNAseq of rat CFs taken at the fetal, neonatal, and adult stages found that inflammation-related genes, including cytokines, are more highly expressed in adult CFs compared with fetal or neonatal[65]. This contributes to the evidence that adult CFs play a role in the inflammatory process whereas developing CFs do not, at least not to the same extent. Another study of adult vs fetal human cardiac fibroblasts found that fetal CFs are more proliferative and that the composition of ECM-associated proteins expressed was different between the two populations[66]. For example, *ELN* and *ITGA8* were expressed in adult CF but had little

expression in fetal CF. Overall, fetal and neonatal CF tend to be more proliferative and migratory, and they express ECM proteins that are part of the normal physiologic milieu. In contrast, adult CF are thought to be more inflammatory and pro-fibrotic in their ECM protein expression pattern. Additionally, although EpiMT and EndoMT have been described as a source of fibroblasts in the adult heart[67,68], the epithelial cell population expressing markers of EpiMT (*TWIST1* and *SPARC*) was absent in adult heart samples but present in fetal heart samples[69]. In that study, epicardial cells also had a shift from expression of genes associated with angiogenesis and EpiMT toward expression of immune-associated genes as they went from fetal to adult epicardial cells. Given that fetal CF are so different from adult CF in behavior and regulators, understanding genetic predisposition for fibrosis in adults as summarized by Ghazal et al[70] is potentially not applicable for identifying potential genetic drivers for congenital fibrosis. In Supplementary Data 3, we identify several genes that are cardiac fibroblast regulators particularly during development and are likely to be involved in risk of fibrosis with CHD.

Beyond genetic predisposition, cardiac fibrosis is also influenced by environmental factors, such as abnormal intracardiac pressures, diminished oxygen delivery, and surgical incisions that come with having structural heart disease. The link between fibrosis and cyanosis or systemic hypoxia has been debated, and the data are unclear[19]. In fact, one small study found that post-surgical patients (that is, formerly cyanotic now with normal oxygen saturations) had the highest levels of cardiac fibrosis, even over currently cyanotic patients[71]. This has led to some speculation that relative cyanosis is protective against fibrosis but the surgical procedure itself conveys some risk for cardiac fibrosis[72].

It's also important to note that patchy cardiac ischemia even in the absence of systemic cyanosis might account for some diffuse replacement

fibrosis seen even remote to the surgical sites[23]. Age, sex, elevated intraventricular pressures, and volume overload have also been linked to cardiac fibrosis[23,72–74]. Additionally, maternal factors, such as autoimmune conditions or maternal diabetes can also lead to fibroblast activation, increased ECM production, and pathological fibrosis during development[75]. Genetic factors likely play a role in predisposing some patients to a higher risk of cardiac fibrosis in response to these environmental factors. A recent study performed GWAS in adult patients with diabetes and correlated genes with amount of fibrosis on cardiac MRI. These include genes related to glucose transport (*SLC2A12*), iron homeostasis (*HFE, TMPRSS6*), tissue repair (*ADAMTSL1, VEGFC*), oxidative stress (*SOD2*), cardiac hypertrophy (*MYH7B*) and calcium signaling (*CAMK2D*)[21]. These findings highlight the range of genes which potentially translate microvascular changes into fibrosis, although the relevance of these genes to fibrosis in CHD is unclear. Below we highlight patterns of pathologic fibrosis in specific forms of CHD.

### Endocardial fibroelastosis and hypoplastic left heart syndrome

One of the most singular congenital fibrosis phenotypes is EFE which is characterized by a thick rind of elastin-rich ECM deposition in the endocardium. This phenotype is most frequently seen with obstructive left-sided cardiac defects, such as hypoplastic left heart syndrome (HLHS), but it can be seen on its own as primary EFE[4]. It has been shown that EFE pathology is caused by an increase in EndoMT leading to a larger population of fibroblasts in the endocardium and an increase in expression of elastin, collagen III, and other ECM proteins[76,77]. Several other genes related to EndoMT including *SNAIL, SLUG, TWIST1, FSP1*, and *CDH5* (VE-cadherin) are upregulated[77]. Loss of function variants in genes related to cilia, such as *ALMS1* have been linked to primary EFE[61].

When seen with HLHS, EFE likely develops around the second or third trimesters[7]. A rat study showed that aortic valve regurgitation in an HLHS model increased EFE over an HLHS model without this volume overload implying the distention of the LV increases risk for EFE[78]. A scRNA-seq study using samples from patients with CHD and controls revealed that two subpopulations of fibroblasts that highly express genes related to fibroblast activation (*TWIST2, ADAMTS4, FGF7, FAP, POSTN*) are enriched in HLHS hearts[3]. It also showed that fibroblasts in HLHS hearts, especially those with heart failure, had high levels of *YAP* expression which correlated with an increase in fibrotic tissue[3]. However, the link between YAP activation and fibrosis is unclear as other studies have shown decreased YAP activity to be associated with fibrosis particularly in adult wound healing. Importantly, YAP signaling has a different effect on CF vs CM and is generally downregulated later in development, so the timing and cell type are important context for understanding the effect of YAP levels[45]. Endothelial cells in the scRNA-seq study above also had unique populations based on gene expression, and, in HLHS, endothelial cells with upregulated mechanotransduction and NOTCH signaling genes were more prominent[3]. This might represent endothelial cells primed to undergo EndoMT giving rise to more activated fibroblasts.

Interestingly, one study found less histological fibrosis and lower *COL1A1* and *COL3A1* expression levels in single ventricle patients with decreased ejection fraction[79]. Additionally, miRNA-29b, a micro-RNA involved in down-regulation of fibrosis-related ECM proteins, such as elastin and fibrillin, was increased in failing single ventricle patients. The amount of fibrosis in failing vs not-failing right ventricles in this study was not clearly quantified but has been shown to be similar. This highlights the importance of ECM composition as certain ECM compositions can be associated with poor function and dilation even in the absence of increased ECM deposition.

### Tetralogy of Fallot

Tetralogy of Fallot (ToF) is a right-sided CHD lesion characterized in most cases by diminished outflow from the right ventricle. In one study, biopsies of pediatric heart samples (all less than 12 months old for tetralogy samples) had no significant fibrosis noted[3]. Fibroblasts in this study did have pathological expression profiles, but they were not upregulating pro-fibrotic

pathways as strongly as fibroblasts found in HLHS and DCM[3]. However, older children and adult patients do have evidence of increased fibrosis on cardiac MRI in both the RV and LV[6,72]. Some studies have shown an association between fibrosis and age at initial repair[6,19], while other studies have not[74]. Increased cross-clamp time, higher RV outflow tract gradient (increased obstruction), increased RV pressures, and increased pulmonary regurgitation have all been associated with increased fibrosis in post-surgical patients with ToF[72,74]. This highlights some of the structural and surgical processes that affect fibrosis independently from genetic factors. The role that genetic predisposition plays in ToF patients is yet to be determined.

### Cardiomyopathy

Fibrosis is a major component of the pathology of hypertrophic cardiomyopathy (HCM). Pro-fibrotic markers, such as MMP-9 and miRNA-29a have both been found at higher circulating levels in HCM compared to controls, and HCM patients have patchy areas of increased fibrosis on cardiac MRI[8]. Importantly, fibrosis was previously thought to be a reaction to the altered cardiac morphology and abnormal mechanical stress on the tissue, but data now suggests fibrosis can precede the hypertrophic changes[10]. Indeed, increased fibrosis is associated with worse outcomes including arrhythmias and heart failure[8,9]. A *MYBPC3* mutation leading to a truncated protein (similar to that seen in HCM) in mice did cause an increase in fibrosis and expression of fibrosis related genes, particularly TGFB members[10], which might point to a role for mechanosensing in this pathophysiology of fibrosis in HCM. Gene expression data also hints at abnormal connections and crosstalk between CM, cardiac fibroblasts, and the ECM which potentially play a role in the development of cardiac fibrosis[10].

Dilated cardiomyopathy (DCM) does have significant fibrosis particularly perivascular and mid-wall, although it is often less than is seen in HCM[3,8]. Fibroblasts in samples from DCM patients have upregulated pro-fibrotic pathways as are seen in ToF and HLHS[3]. Patients with DCM and fibrosis have particularly increased circulating procollagen III, but all DCM patients regardless of fibrosis levels had increased circulating markers of fibrosis (eg CTGF, MMP9)[8]. All of this clinical data points to an increased risk for fibrosis in DCM, but the mechanistic cause is still unclear.

LV non-compaction is a rare cardiomyopathy characterized by an abnormal spongy structure to the myocardium generally leading to a poor systolic and diastolic function[80]. Non-compaction is related to increased interstitial fibrosis primarily in the interventricular septum although there have been patients observed with non-compaction and EFE[8,80]. However, the amount of fibrosis observed is generally less than in other types of cardiomyopathies[8].

### General factors that affect risk for fibrosis in CHD

There are several syndromes or predisposing factors that commonly increase the risk of fibrosis in CHD. For example, chromosomal anomalies, particularly trisomy 21, are associated with derangements in expression of many genes and increase the risk for CHD[81]. Many of the genes that are abnormally expressed in trisomy 21 are related to collagen expression or calcium signaling – pathways that are likely to be related to cardiac fibrosis[16]. Ciliopathies, including polycystic kidney disease (*PKD1* or *PKD2* mutation) and primary ciliary dyskinesia (*DNAH11* mutation among others), seem to increase the risk for CHD generally in addition to increasing risk for heterotaxy specifically. Maternal diabetes is associated with cardiac hypertrophy and increased risk for CHD[82]. Emerging evidence is linking abnormal expression of certain genes with risk of developing CHD in the context of maternal diabetes in rat and mouse studies[83,84]. Specifically, many genes related to those outlined in Supplementary Data 2, such as Tgfb1 family members (*Tgfbr1, Bmp4, Bmp10, BAMBI*), calcium related signaling (*Nfatc4*), and drivers of EndoMT (*Twist2*) as well as genes that regulate retinoic acid signaling have all been shown to be dysregulated during cardiogenesis in the presence of maternal diabetes. This matches the increase in cardiac fibrosis observed in fetal mice in the context of maternal diabetes. However, there is a lack of such studies in humans, and no data on fibrosis in that context.

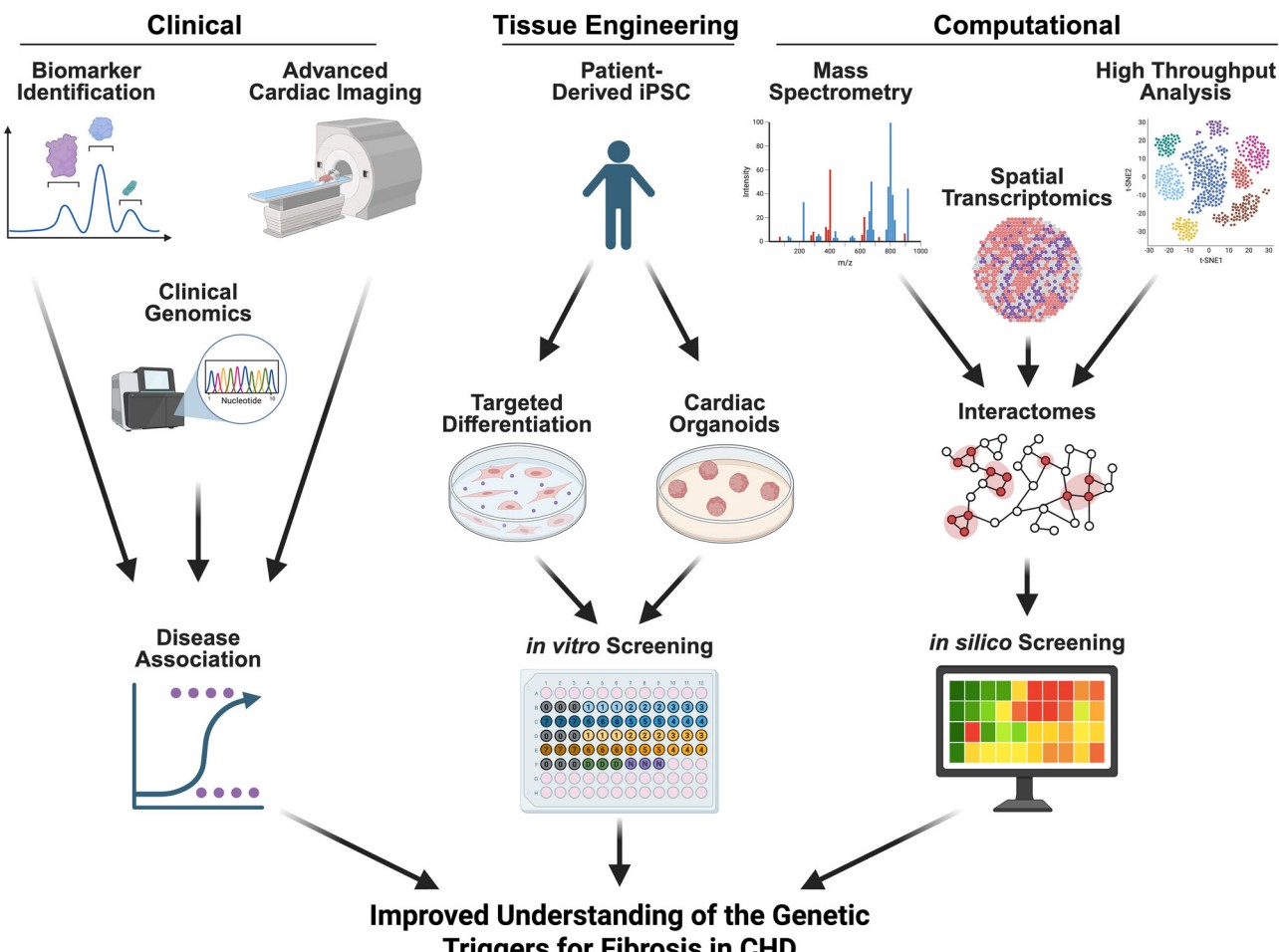

**Fig. 4 | Future directions in the genetics of fibrosis in congenital heart disease (CHD).** Shown are new and improving techniques in clinical characterization, tissue engineering, and computational methods and the major experimental studies they will enable in order to further investigate the relationship between genetic predisposition and fibrosis in CHD. *Created in BioRender. Zeigler, A. (2026)* https://BioRender.com/*v0kdqnn*.

Cardiac defects that have a systemic RV circulation, such as truncus arteriosus or transposition of the great arteries, tend to develop progressive fibrosis that is worse the later the repair. Patients with transposition who received the atrial switch operation rather than the arterial switch (therefore continuing to have an RV supplying systemic circulation) had high levels of interstitial cardiac fibrosis measured on cardiac MRI[6]. This is likely related to abnormal mechanical stress on the RV as the pressures needed to supply systemic blood flow are higher than a normal RV would experience supplying pulmonary blood flow. After the development of RV hypertrophy from chronic pressure overload in patients with a systemic RV patchy fibrosis is characteristic, likely related to inadequate blood supply[19]. Patients with transposition who get an arterial switch operation end up with a systemic LV, and, in those patients, fibrosis is uncommon and more likely to be related to periods of ischemia, such as with abnormal coronary flow[19].

Other lesions with increased pressure load on the LV, such as aortic stenosis have increased fibrosis, and in one study the pattern was more mid-wall fibrosis like that seen in DCM[6]. However, a study of younger patients found that fibrosis on imaging did not correlate with end-diastolic pressures or extent of aortic stenosis which indicates further investigation may find a genetic predisposition for fibrosis in these cases[6].

Lesions that involve chronic cyanosis also lead to RV fibrosis[19,61]. However, the link is not totally clear because a small study found that patients with cyanotic heart disease (eg: pulmonary atresia with VSD, truncus arteriosus, or atrioventricular septal defects) who were corrected had higher levels of fibrosis than those who were not[71]. In this study, it seems that surgical incision or use of cardiopulmonary bypass or both are likely stronger contributors to fibrosis. Patients in this study were young (median age 33 or less), meaning ongoing cyanosis could have a more profound effect.

## Future directions

Newer technologies that have been applied to understanding CHD and other birth defects could be used to investigate the genetic risk factors for fibrosis in CHD. While there are many studies that relate the structural phenotype of CHD to different genetic mutations or altered gene expression patterns and signaling pathways[11,60,63,85], there were no large scale studies we could find that perform a similar analysis for congenital or infantile cardiac fibrosis[9,21]. However, there are new technologies that could open this field up for more in-depth study. These clinical, tissue engineering, and computational advancements are summarized in Fig. 4.

With the improvement in circulating biomarkers and advanced cardiac imaging now we are able to quantify cardiac fibrosis non-invasively, larger studies linking genetic mutations to clinical evidence of cardiac fibrosis are more feasible[21,23,29]. In fact, one study by Nauffal et al was able to explore the link between a variety of disease states and lifestyle factors with cardiac fibrosis using machine learning approaches to analyze cardiac MRI[21]. They also performed GWAS on over 400,000 patients and linked the genes *ADAMTSL1* and *SLC2A12* with fibrosis in adults among other genes highlighting the increasing utility of clinical genomic studies.

Tissue engineering techniques have also been leveraged to better understand cardiac development. The use of patient-derived iPSC's has grown with the ability to differentiate cells into epicardial cells, CF, or CM. A

review by Lin et al highlighted several studies using patient-derived iPSCs to uncover the mechanism behind specific gene variants associated with CHD[27]. One study used iPSC-derived endothelial cells from patients with HLHS and a control and found multiple potentially causal abnormalities including altered NOTCH signaling and integrin expression[86]. The development of cardiac organoids, where some combination of CM, CF, epicardial cells, and endothelial cells organize in a 3D model of cardiac tissue, has allowed for detailed and controlled study of the interplay between different cell types. One study using human embryonic stem cells cultured in Matrigel captured the organization of these cells into a structure with a distinct layer of CM limned by endocardial cells as well as collections of cells that appeared to be foregut endoderm[87]. Hofbauer et al used human iPSC to develop cardiac organoids that formed chamber-like structures, and used them to examine the role for different genes in cardiogenesis as well as the effect of injury[88]. This study highlighted the utility of this technique in examining the effect of genetic variants on normal development and response to injury. Other studies have outlined the utility of iPSC and cardiac organoids in screening for therapeutic options or in creating new biomaterials for surgical correction[26,89,90]. This technology will allow for a more mechanistic study of potential genetic causes for CHD particularly how they relate to ECM development and cardiomyocyte-fibroblast interactions.

Additionally, advances in computation and machine learning have allowed for extraction of information from large datasets. Large-scale interactome development could identify a network of proteins that convey risk in a multigenic fashion[62,91]. RNAseq studies at the single cell resolution have already been used to map different expression profiles for different CHD phenotypes[3,86], and some studies have examined altered expression profiles under external stresses, such as abnormal flow patterns[76]. Spatial transcriptomics has also emerged as a way to study expression patterns on the cellular level while maintaining the spatial organization[92,93]. This is particularly useful in studying fibrosis where the location of the fibroblasts or endothelial cells in relation to increased ECM can give more information about what a pro-fibrotic expression pattern looks like. Large-scale transcriptome datasets have been used to create co-expression networks which have fueled hypotheses for further mechanistic study. High-throughput proteomics studies using mass spectrometry or NMR have been used to develop protein-protein interaction networks[94], some of which now exist as publicly-available databases, such as string (https://string-db.org/). These interaction networks can be the basis of computational models where the effect of therapeutics and genetic variation can be virtually screened[95,96].

Because fibrosis is so heterogenous and multifactorial, a good framework for using these newer technologies to find genetic modifiers of cardiac fibrosis in CHD is to think about genes in the context of different developmental processes. Figure 1 outlines the various stages of development. Genes associated with these stages are likely to cause altered risk for fibrosis particularly in the context of CHD where the risk of inflammation, infarction, and (surgical) injury are higher. Using integrated high-throughput patient-derived omics and molecular genetic experiments to link genes to these different stages of fibroblast development could provide a more mechanistic understanding of genetic risk for fibrosis in CHD and pave the way for fibrosis-targeted therapies. Ultimately, this research to better understand the genetic predisposition for cardiac fibrosis will potentially allow for the design of better anti-fibrotic therapies and earlier identification of CHD patients at risk of developing cardiac fibrosis.

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

## Author contributions

A.C.Z. Manuscript conception, design and writing, and generating figures. M.T. Manuscript conception, design, writing and editing, and managing fund. All authors have read and agreed to the published version of the manuscript.

## Funding

ACZ discloses support for the research of this work from the UCLA Intercampus Medical Genetics Research Training Program: USHHS Ruth L. Kirschstein Institutional National Service Award # T32GM008243. MT discloses support for this work from NIH/1R01 HL153853, the Department of Defense Congressionally Directed Medical Research Programs (CDMRP)-DMD-Idea Development Award HT9425-25-1-0763 Project Number: MD240028, and the UCLA Academic Senate Faculty Research Fund.

## Competing interests

No competing Interests. The authors declare that the research was conducted in the absence of any commercial or financial relationships that could be construed as a potential conflict of interest.
