## [Transparent Peer Review file · Communications Biology]

Genetic Drivers of Congenital Cardiac Fibrosis

Corresponding Author: Professor Marlin Touma

Version 0:

Reviewer comments:

Reviewer #1

(Remarks to the Author)

While the authors summarize the current research on fibroblasts in congenital heart fibrosis, several important aspects require further development. In particular, the section on cilia is underdeveloped and lacks mechanistic depth. The authors should expand this section by adding a schematic figure and a more detailed discussion of how ciliary defects influence cardiac development and fibrosis, including the relevant cellular and molecular pathways. Furthermore, the manuscript does not adequately address whether the mechanisms governing fibroblast activation and fibrosis progression in the developing heart differ from those in the adult heart. Clarification and comparison of these developmental versus adult mechanisms would substantially strengthen the manuscript.

Reviewer #2

(Remarks to the Author)

The review by Zeigler et al. focused on describing the genetic drivers of congenital cardiac fibrosis. Authors describe the reasons for worse outcomes in CHD patients with fibrosis, including that cardiac fibroblasts play an important role in normal cardiac development and cell differentiation and that different CHD phenotypes have different patterns of fibrosis that may be mediated from different genetic mutations. Authors also describe the use of novel technologies such as scRNA-seq, patient-derived iPSCs, cardiac imaging, and fibrosis-associated biomarkers as integral for improving identification of drivers of cardiac fibrosis in the future.

This is a well-written and comprehensive review. Some suggestions to improve the review include needing additional figures and charts to clarify the reviewed material:

- 1) Discussion of use of iPSC-derived cardiac organoids to further the understanding of development of fibrosis in CHD is not acknowledged or introduced.
- 2) There should be a figure showing the development of the heart and where/what mutations lead to fibroblast-related CHDs, including both genetic and environmental factors.
- 3) Table 1 would benefit from having a column indicating timing of expression; for example, when are the various markers induced, embryonically or in adults, and by what or in response to what?
- 4) There should be a figure ascribing the role of cilia in fibroblasts.
- 5) Table 2 should also include the genes that are induced as a consequence of environmental and other disease factors (with a column ascribing as such).
- 6) A figure showing the mechanisms and causal genes of each of the indicated CHDs should be generated: hypoplastic left heart syndrome, Tetralogy of Fallot, and HCM should be included.
- 7) A chart of the other CHD lesions should also be included.
- 8) More detailed future studies sections is needed, including addition of a figure for the section that describes each new technology (imaging, sc-RNA-seq, and ipscs, etc).
- 9) Figure 1 is confusing and not detailed enough. What leads to normal vs abnormal development? What is reactive fibrosis? What is the timing for all of this? It would be better to delineate these in the above sections with figures within each section as opposed to trying to summarize as shown here.

Version 1:

Reviewer comments:

Reviewer #1

(Remarks to the Author)
My comments were addresses.

Reviewer #2

(Remarks to the Author)
Authors have adequately addressed previous concerns.

Responses to Reviewers,

We are pleased to submit our revised review manuscript [COMMSBIO-25-10318A]
Title: Genetic Drivers of Congenital Cardiac Fibrosis.

We appreciate the thoughtful comments and suggested revisions from the reviewers. We have added new figures in order to clarify the cardiac development process, the types of fibrosis, ciliary function, and the future directions in this area of research. We also added to the mechanistic discussion of ciliary function as well as a more detailed description of the differences between adult and neonatal cardiac fibroblasts. Finally, we added a table outlining genes strongly associated with structural congenital heart disease. We feel these additions have greatly added to the understanding of fibrosis particularly within the context of current knowledge of the pathogenesis of CHD. Further, the important “**Key Points**” of this review article are provided after the revised (shortened) **Abstract** aiming to draw the readers’ attention and focus. We summarize our responses to the reviewers’ comments below:

Reviewer 1 comments:

1) While the authors summarize the current research on fibroblasts in congenital heart fibrosis, several important aspects require further development. In particular, the section on cilia is underdeveloped and lacks mechanistic depth. The authors should expand this section by adding a schematic figure and a more detailed discussion of how ciliary defects influence cardiac development and fibrosis, including the relevant cellular and molecular pathways.

Response. We agree that the discussion of ciliary function and ciliary defects was not sufficiently detailed. We have re-written **section 2.2** in order to incorporate some of this important mechanistic detail. Specifically, we have included further discussion of the basics of cilia structure and their roles during development (namely left-right patterning, migration, and coordination of signaling processes). We also included a new figure (**Figure 2**) that details the basic structure of cilia and several important signaling receptors representing molecular pathways that cluster there. Further, in this figure, we highlighted structural and signaling genes with a connection to cardiac development and/or CHD. We feel this expansion of our discussion on cilia has added more depth to the overall context for cilia and their important roles in cardiac development and fibrosis.

2) Furthermore, the manuscript does not adequately address whether the mechanisms governing fibroblast activation and fibrosis progression in the developing heart differ from those in the adult heart. Clarification and comparison of these developmental versus adult mechanisms would substantially strengthen the manuscript.

Response. We did not address how fibrosis activation in the developing / neonatal heart differs from the adult heart primarily in an effort to avoid confusing signaling prominent in adult fibrosis with the signaling active during cardiac development. However, we agree that some discussion of adult fibroblasts is warranted since a majority of our understanding of cardiac fibrosis comes from adult studies. Therefore, we added a paragraph in **section 3.1 [Paragraph 2]** regarding some of the key differences in signaling pathways, and we agree this adds important distinction from the many genes associated with adult heart failure or post-MI fibrosis that may not be relevant in development. Additionally, we added a column in **Table 1** that indicates the timing of marker becoming expressed including whether it continues to be expressed in adults, and we added a column in **Table 3** (formerly table 2) that indicates the known context of that gene being expressed (some of which are more relevant in adults like ischemia / reperfusion). Altogether, we feel these changes address some of the major differences and similarities between adult and developmental fibroblasts.

Reviewer 2 comments:

1) Discussion of use of iPSC-derived cardiac organoids to further the understanding of development of fibrosis in CHD is not acknowledged or introduced.

Response. We had a short sentence on the use of iPSC-derived cardiac organoids in our future directions, but we agree that further detail would add to the future directions in **section 4**. To that end, we rewrote section 4, and added more detailed background on the advancements made in cardiac organoids [**Paragraph 3**] while making a clear link to these exciting studies and the future studies needed to characterize the genetics of fibrosis.

2) There should be a figure showing the development of the heart and where/what mutations lead to fibroblast-related CHDs, including both genetic and environmental factors.

Response. We agree that our description of normal cardiac development would be improved with an explanatory figure. We created a figure showing the major events in cardiac development and highlighted the major roles fibroblasts play in those events. We also outlined key genes known to be important for normal cardiac development – many of which are also linked to CHD pathogenesis. To our knowledge, there is not as clear of a link between specific environmental factors and alterations in fibrosis genes during these major events of cardiac development. For that reason, we limited the discussion of environmental factors to **Table 3 (formerly table 2)** where we added a column indicating the contexts, including environmental contexts, within which these genes have different expression.

3) Table 1 would benefit from having a column indicating timing of expression; for example, when are the various markers induced, embryonically or in adults, and by what or in response to what?

Response. We agree that timing and context of expression were important additions to **Table 1**. We included a column indicating timing and, when known and relevant, specific triggers for each marker.

4) There should be a figure ascribing the role of cilia in fibroblasts.

Response. This was also mentioned by reviewer 1, and we agree such a figure is important for highlighting the ciliary function during development. We made new **Figure 2** to address this concern. The diagram includes structural components and multiple receptors representing signaling pathways that cluster in the cilium. The downstream functions of those signaling pathways are indicated highlighting the diverse roles cilia perform in cardiac development and relation to cardiac fibrosis.

5) Table 2 should also include the genes that are induced as a consequence of environmental and other disease factors (with a column ascribing as such).

Response. We added a column to **Table 3** (formerly table 2) indicating the context under which the expression of each gene is altered. We also added two genes recently tied to CHD in the context of a diabetic mother and to fibrosis in the context of diabetes, which we felt was an important environmental risk factor for CHD in general. Additionally, the first paragraph of **section 3.5** discusses other disease states including genetic syndromes and gestational diabetes as triggers for CHD. Where possible, specific genes are indicated in relation to those disease states.

6) A figure showing the mechanisms and causal genes of each of the indicated CHDs should be generated: hypoplastic left heart syndrome, Tetralogy of Fallot, and HCM should be included.

Response. Rather than a figure, we felt a table was more useful to address this important point, primarily because of the polygenic cause for most CHD with only few confirmed single causal genes in non-syndromic CHD. Additionally, many larger studies looking for causal genes necessarily lump different CHD phenotypes

together because of the small number of patients, and this makes it difficult to link a gene to a specific phenotype rather than just CHD in general. There is a long list of genes associated with varying degrees of certainty with CHD which has been reviewed many times, and we included a more detailed description of the major categories of these genes as well as a more explicit citing of recent comprehensive reviews on the subject that the readers can refer to. However, we agree that a more detailed discussion of genes with a strong link to CHD is necessary in this review – even those genes that have no clear link to fibrosis risk as of yet. For this reason, we included a new table (now **Table 2**) highlighting a few genes with a clear link to CHD as well as the likely underlying mechanism when that is known. The first paragraph of **Section 3** discusses some of these specific genes in more detail and their possible relationship to cardiac fibrosis. Additionally, **Figure 1** showing the overall developmental process highlights genes important for normal development, and many of those genes have also been linked to CHD development. We further discuss the implication of specific mutations during these stages of development in the first paragraph of **Section 2**. None of these are comprehensive lists of potential causal genes, and they are not intended to be. Instead, they are intended to highlight major pathways that are linked to CHD in humans in order to discuss any potential tie to fibrosis. We have cited key reviews and studies that expanded on this list for our readers who desire a more comprehensive list of genes.

7) A chart of the other CHD lesions should also be included.

We agree that the section on “other CHD lesions” was sparse and not clearly targeted to specific CHD lesions. For this reason, we rewrote **Section 3.5** to instead address general factors that can affect fibrosis risk in congenital heart disease. Rather than discussing all of the other types of CHD lesions which have not been as clearly related to fibrosis, we centered the discussion around environmental factors that can cause fibrosis in many different CHD pathologies, including maternal diabetes, pressure overload, and prolonged cyanosis. Additionally, we created a new **Table 2** that outlines specific genes related to CHD including a column indicating which CHD pathology has been linked to a mutation in that gene. We feel these changes highlight the point we were trying to communicate which is that there are other factors that convey risk for fibrosis even outside of the fibrosis patterns which are strongly associated with the specific CHD pathologies outlined in their own sections. We felt further discussion of all CHD lesions would be outside the scope of this review partly because of the long list of specific CHD pathologies and partly because most CHD types are not strongly linked with fibrosis in the absence of the risk factors described in this new section.

8) More detailed future studies section is needed, including addition of a figure for the section that describes each new technology (imaging, sc-RNA-seq, and ipscs, etc).

Response. We rewrote **Section 4** on future directions to have more detail regarding the new technologies available and how those could be linked to the study of congenital fibrosis. Additionally, we added **Figure 4** which outlines how these techniques could be used for further investigation. We feel this adds some depth to the discussion of these exciting advancements.

9) Figure 1 is confusing and not detailed enough. What leads to normal vs abnormal development? What is reactive fibrosis? What is the timing for all of this? It would be better to delineate these in the above sections with figures within each section as opposed to trying to summarize as shown here.

Response. We appreciate the feedback that this figure is confusing. To address this, we have separated this into three figures: 1) **Figure 1** outlining the general overview of cardiac development at both the organ and cellular fibroblast level with some genes and major environmental factors delineated along the timeline. 2) **Figure 2** detailing ciliary structure and function in cardiac development with genes and mechanisms highlighted. 3) **Figure 3** outlining the pathogenesis of fibrosis with specific genes highlighted.

To address the question of what is normal vs abnormal development, we added to the first paragraph in **section 2** that details normal development and the implication of abnormalities in these stages of development. This pairs nicely with **Figure 1** to clarify normal development. Throughout the rest of the manuscript including the details highlighted in the new **Table 2** are details regarding when abnormal development leads to CHD and any associated genetic mutations.

We appreciate your question regarding reactive fibrosis. Because this is a somewhat vague term used to describe fibrosis in response to inflammation generally and can encompass interstitial and replacement fibrosis, we have removed this term from the manuscript. Instead, we focused on the distinct patterns of fibrosis including interstitial, replacement, and EFE which are visually depicted in a new **Figure 3** and are discussed in detail in the first paragraph of **Section 3.1**.

The timing of different stages of development are highlighted in **Figure 1**. The timing of different fibrosis associated signaling pathways is also discussed in **Section 3.1 [Paragraph 1]**.